# Comprehensive Draft Genome Analyses of Three Rockfishes (Scorpaeniformes, *Sebastiscus*) via Genome Survey Sequencing

**Chenghao Jia** [1]**, Tianyan Yang** [2] **, Takashi Yanagimoto** [3] **and Tianxiang Gao** [2,*]

[1] State Key Laboratory of Marine Resource Utilization in South China Sea, College of Ecology and Environment, Hainan University, Haikou 570228, China; xicheng121@163.com
[2] Fishery College, Zhejiang Ocean University, Zhoushan 316022, China; hellojelly1130@163.com
[3] Fisheries Resources Institute, Japan Fisheries Research and Education Agency, Kanagawa 236-8648, Japan; yanagimo@fra.affrc.go.jp
[*] Correspondence: gaotianxiang0611@163.com; Tel.: +86-0580-255-6416

**Abstract:** *Sebastiscus* species, marine rockfishes, are of essential economic value. However, the genomic data of this genus is lacking and incomplete. Here, whole genome sequencing of all species of *Sebastiscus* was conducted to provide fundamental genomic information. The genome sizes were estimated to be 802.49 Mb (*S. albofasciatus*), 786.79 Mb (*S. tertius*), and 776.00 Mb (*S. marmoratus*) by using k-mer analyses. The draft genome sequences were initially assembled, and genome-wide microsatellite motifs were identified. The heterozygosity, repeat ratios, and numbers of microsatellite motifs all suggested possibly that *S. tertius* is more closely related to *S. albofasciatus* than *S. marmoratus* at the genetic level. Moreover, the complete mitochondrial genome sequences were assembled from the whole genome data and the phylogenetic analyses genetically supported the validation of *Sebastiscus* species. This study provides an important genome resource for further studies of *Sebastiscus* species.

**Keywords:** genome size; *Sebastiscus*; microsatellite; phylogenetic analysis; genome survey sequencing

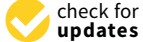



## 1. Introduction

The genus *Sebastiscus* is an interesting and commercial rockfish that mainly distributes in the Western Pacific [1]. Despite their diversity, abundance, and economic importance, our understanding of the relationships within the genus remains limited. Three species have been recognized from *Sebastiscus* viz., *Sebastiscus albofasciatus* (Lacepede, 1802), *Sebastiscus marmoratus* (Cuvier and Valenciennes, 1829), and *Sebastiscus tertius* (Barsukov and Chen, 1978) [2–4]. Additionally, *S. tertius* was generally misidentified as *S. marmoratus* in the Chinese mainland coastal waters in the last decades because of the similar morphological characteristics. It is also worth noting that *S. marmoratus* is widely distributed in the northwestern Pacific Ocean and the remaining species are likely to be confined to the warm waters of East Asia and Indonesia [5,6]. Recently, combined with morphological and DNA-barcoding approaches, this species was determined in the Chinese mainland coastal waters as a new record species [7]. Although the external morphology of the *S. tertius* and *S. marmoratus* is more similar than *S. albofasciatus*, recent studies have shown that *S. tertius* is more closely related to *S. albofasciatus* than *S. marmoratus* at the genetic level [8]. As a result, it further reveals that the evolutionary relationship of *Sebastiscus* fishes is necessary to make a deeper comparison and discussion of their genes. However, the limited genetic information and the lack of the genomic data of *Sebastiscus* have impeded the relevant evolutionary and genomic studies of this genus.

As the next-generation, high-throughput sequencing (NGS) has been developing rapidly over recent decades, the whole genome sequencing (WGS) and various molecular genetic tools have become efficient strategies for generating genomic resources and widely applied in genomic and evolutionary studies of marine fish species [9–13]. To further

compare the genomes of different *Sebastiscus* species, complete genome data were obtained by NGS and used to assemble the genome, estimate genome size, identify simple sequence repeats (SSRs), and extract the mitogenome. These data will be the basis of a fundamental genomic resource for extending our current knowledge of *Sebastiscus* genome organization and difference. In addition, these data also provide a foundation for future genomic studies of *Sebastiscus* species.

## 2. Results

### 2.1. Genome Sequencing, K-Mer Analysis, and Genome Assembly

A total of 40.14 Gb (*S. albofasciatus*, sequencing depth ~50×), 41.93 Gb (*S. tertius*, sequencing depth ~53×), and 87.55 Gb (*S. marmoratus*, sequencing depth ~112×) clean data were generated by whole genome sequencing. The amount of clean data, Q20, Q30, and GC content of clean data are shown in Table 1 and Figure 1. After a quality control and data filtering analysis, the clean data were used for k-mer analysis. The 19-mer frequency distribution derived from the sequencing reads is plotted in Figure 2. The k-mer analyses showed that the peaks of 19-mer distribution of three rockfishes were at 21×, 22×, and 83×, respectively (Table 2; Figure 2). Additionally, the estimated genome sizes of *S. albofasciatus*, *S. tertius*, and *S. marmoratus* were 802.49 Mb, 786.79 Mb, and 776.00 Mb by calculation, respectively. The heterozygosity ratio and repeat ratio are also shown in Table 2.

**Table 1.** Quality control information of sequencing data.

| Species | Clean Data [1] (bp) | Q20 [2] | Q30 [3] | GC Content (%) |
|---|---|---|---|---|
| *S. albofasciatus* | 40,142,938,530 | 97.03 | 92.46 | 40.64 |
| *S. tertius* | 41,927,858,708 | 96.99 | 92.31 | 40.77 |
| *S. marmoratus* | 87,553,495,520 | 96.08 | 90.72 | 43.00 |

[1] Clean data: data obtained after deletion of low quality reads. [2] Q20: the ratio of data with accuracy above 99% in total data. [3] Q30: the ratio of data with accuracy above 99.9% in total data.

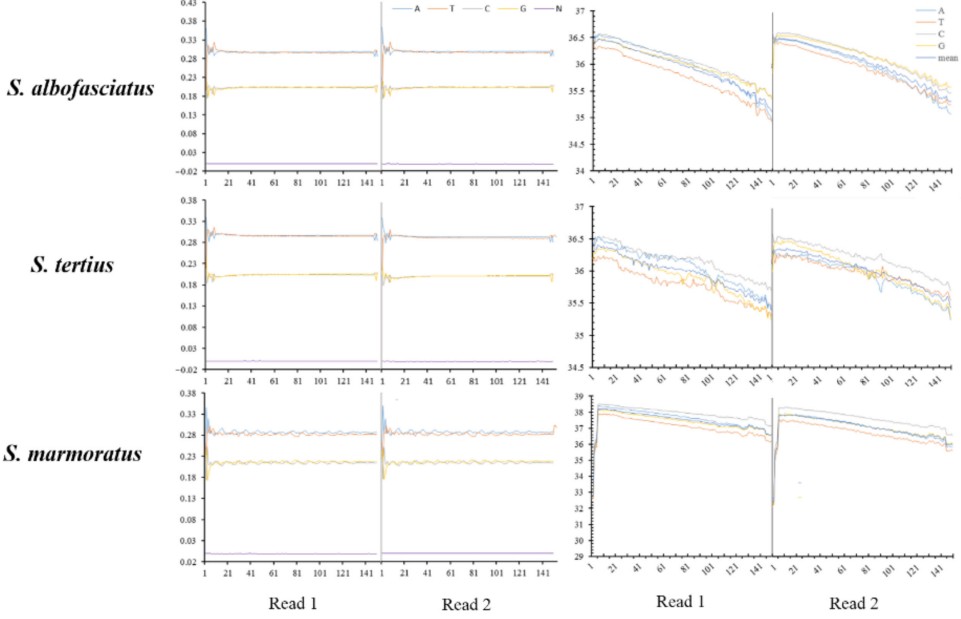

**Figure 1.** Distribution figure of sequencing quality and GC content of clean data.

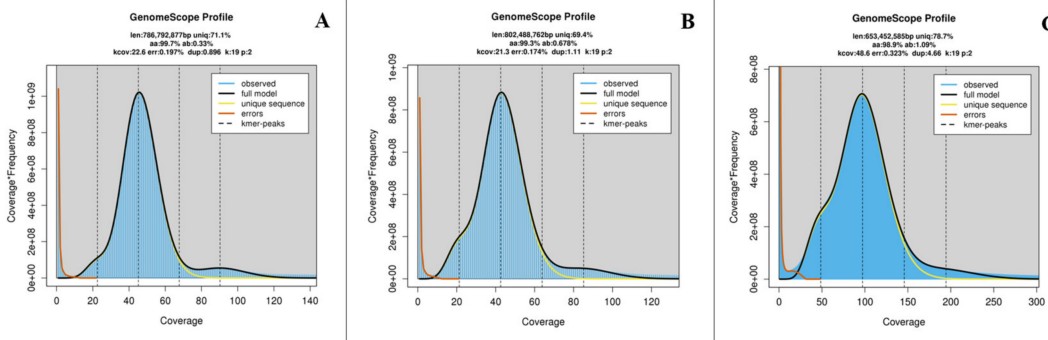

**Figure 2.** K-mer (19-mer) analysis for estimating the genome size of *S. albofasciatus* (**A**), *S. tertius* (**B**), and *S. marmoratus* (**C**). The *X*-axis is depth and the *Y*-axis is the proportion that represents the frequency at that depth.

**Table 2.** Statistics of 17-mer analysis.

| Species | K-Mer Depth | Genome Size (Mb) | Heterozygous Ratio (%) | Repeat Ratio (%) |
| --- | --- | --- | --- | --- |
| *S. albofasciatus* | 21 | 802.49 | 0.68 | 30.59 |
| *S. tertius* | 22 | 786.79 | 0.33 | 28.93 |
| *S. marmoratus* | 83 | 776.00 | 0.94 | 39.94 |

The filtered clean data was used for the draft genome assembling. The total length, total number of sequences, max length of sequences, length of N50, and length of N90 of three rockfishes are shown in Table 3 at the contig and scaffold levels. By comparison, the N50 length and N90 length of sequences of *S. marmoratus* genome were smaller than *S. albofasciatus* and *S. tertius*, while the lengths of max of *S. marmoratus* genome were relatively larger than those of other *Sebastiscus*, especially for the length of contig.

**Table 3.** Statistics of assembled draft genome sequences.

| Species | | Total Length (bp) | Total Number | Max Length (bp) | N50 Length (bp) | N90 Length (bp) |
| --- | --- | --- | --- | --- | --- | --- |
| *S. albofasciatus* | Contig | 528,030,961 | 1,342,756 | 12,292 | 628 | 157 |
| | Scaffold | 544,202,783 | 322,389 | 24,591 | 2257 | 595 |
| *S. tertius* | Contig | 541,266,830 | 1,285,695 | 13,369 | 715 | 162 |
| | Scaffold | 537,925,372 | 292,865 | 28,906 | 2462 | 608 |
| *S. marmoratus* | Contig | 545,203,705 | 1,519,309 | 31,830 | 569 | 142 |
| | Scaffold | 442,737,491 | 301,745 | 31,807 | 1677 | 265 |

### 2.2. Identification of Microsatellite Motifs

A total of 346,510, 382,140, and 319,533 microsatellite motifs were identified for three rockfishes based on the assembled draft genome sequences (Table 4). The microsatellite distribution frequencies in *S. albofasciatus*, *S. tertius*, and *S. marmoratus* genomes were estimated to be about 556.8, 602.4, and 516.9 microsatellites per Mb. In the three genomes, the microsatellites motif types of *S. albofasciatus* included 36.45% mononucleotide, 50.30% dinucleotide, 11.78% trinucleotide, 1.33% tetranucleotide, 0.13% pentanucleotide, 0.02% hexanucleotide repeats; and *S. tertius* had 37.93% mononucleotide, 49.12% dinucleotide, 11.48% trinucleotide, 1.34% tetranucleotide, 0.11% pentanucleotide, and 0.01% hexanucleotide repeats; while the frequencies were 37.43%, 50.12%, 11.31%, 0.10%, and 0.02% for mono-, di-, tri-, tetra-, penta-, hexanucleotide repeats in the *S. marmoratus* genome (Figure 3). Among the microsatellite motif profiles of three rockfish genomes, the dinucleotide repeat motifs were similar to each other and the AC/CA/GT/TG repeats were the most abundant, accounting for about 67.00% (Figure 4A–C). Of the trinucleotide repeat motifs, the repeat times that were more than 1000 included 14, 16, 13 different microsatellite motifs, respectively, accounting for more than 50.00% (Figure 4D–F). However, the tetranucleotide repeat motifs showed significant differences: the genomes of *S. albofasciatus* and

*S. tertius* included about 25 tetranucleotide repeat motifs that were more than 50 repetitions, but *S. marmoratus* only had 3 (Figure 4G–I).

**Table 4.** Microsatellite motif types detected in this study.

|  | *S. albofasciatus* | *S. tertius* | *S. marmoratus* |
|---|---|---|---|
| Total number of sequences examined | 533,972 | 531,582 | 880,807 |
| Total size of examined sequences (bp) | 622,320,988 | 634,375,421 | 618,186,550 |
| Total number of identified SSRs | 346,510 | 382,140 | 319,533 |
| Number of SSR containing sequences | 188,050 | 207,292 | 210,193 |
| Number of sequences containing more than 1 SSR | 78,166 | 87,618 | 61,681 |
| Number of SSRs present in compound formation | 31,111 | 31,522 | 21,646 |

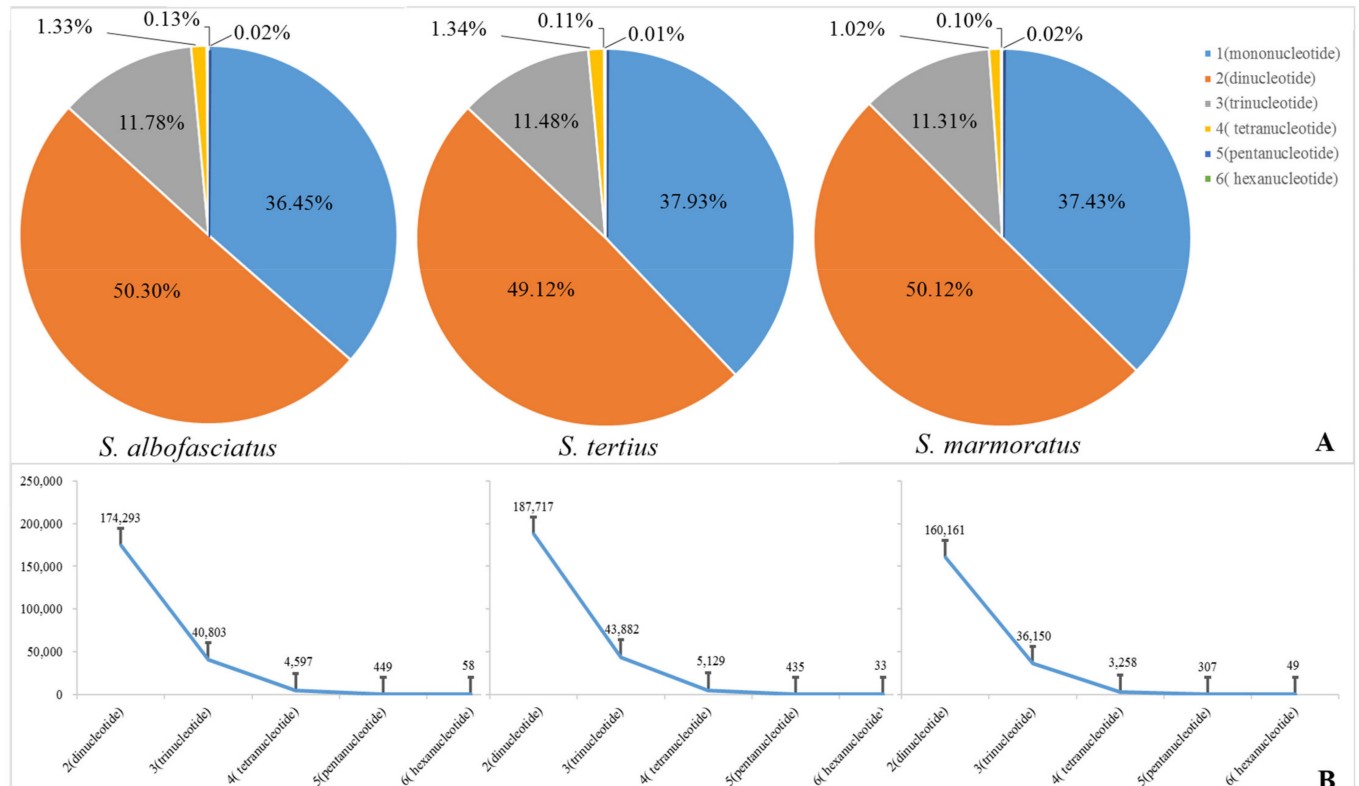

**Figure 3.** (**A**) Frequency of identified microsatellite motif types. (**B**) Quantity of identified microsatellite motifs.

### 2.3. Mitogenome Assembly and Phylogenetic Analysis

The complete mitochondrial genomes of *S. albofasciatus* (Accession no. MZ902351), *S. tertius* (Accession no. MZ902352), and *S. marmoratus* (Accession no. MZ902353) in GenBank were 16,790, 16,797, and 17,208 bp in length, respectively (Figure 5). The size variation of the three mitogenomes was mainly caused by the differences in the lengths of the non-coding regions. The mitogenome of three *Sebastiscus* all contained the typical 37 genes (13 PCGs, 22 tRNAs, and 2 rRNAs), 1 control region, and 1 L-strand replication region (O$_L$). Most mitochondrial genes were encoded on the H-strand, except for ND6 and eight tRNA (Glu, Ala, Asn, Cys, Tyr, Ser-UCN, Gln, and Pro) genes that were encoded on the L-strand. The nucleotide composition of *S. albofasciatus*, *S. tertius*, and *S. marmoratus* mitogenomes had a higher A + T bias of 54.73%, 54.69%, and 54.97%, respectively, and both showed positive AT-skew and negative GC-skew (Figure 6; Table S1).

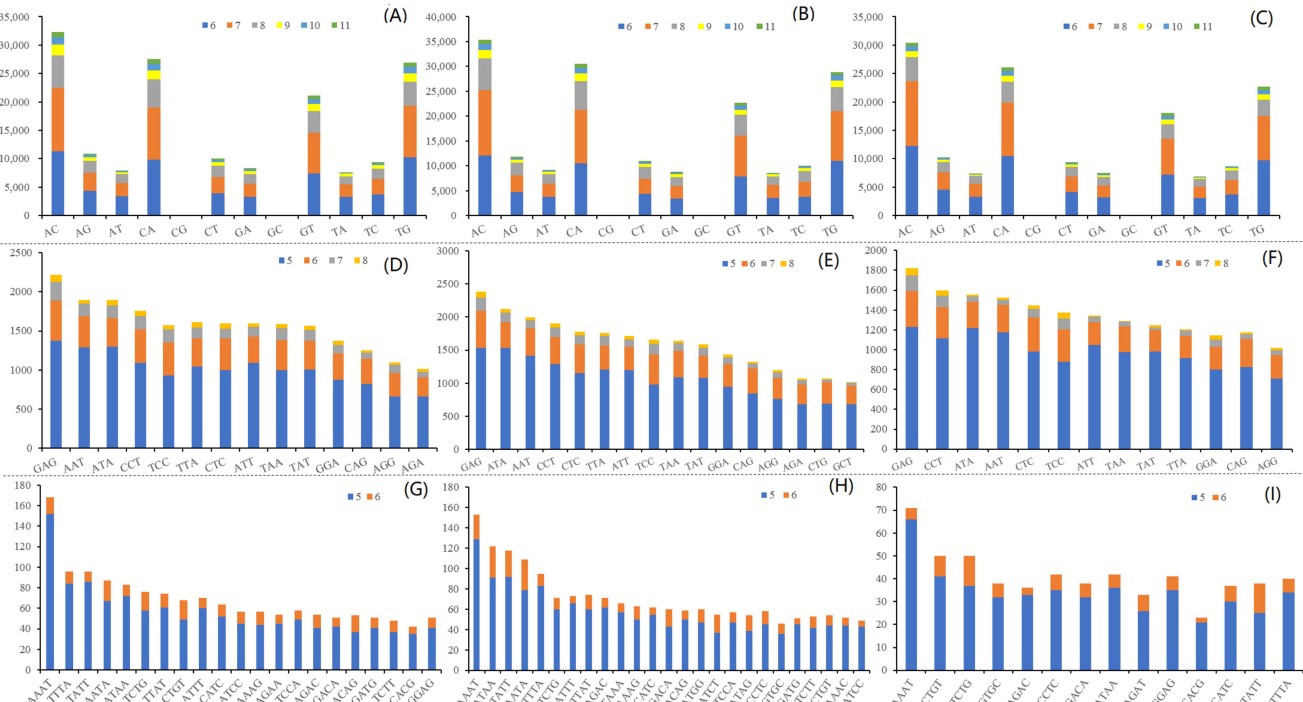

**Figure 4.** The distribution of identified microsatellite motifs. (**A–C**) Frequency of different dinucleotide microsatellite motifs. (**D–F**) Frequency of different trinucleotide microsatellite motifs (>1000). (**G–I**) Frequency of different tetranucleotide microsatellite motifs (>40). The squares in different colors with particular numbers represent different repeat times.

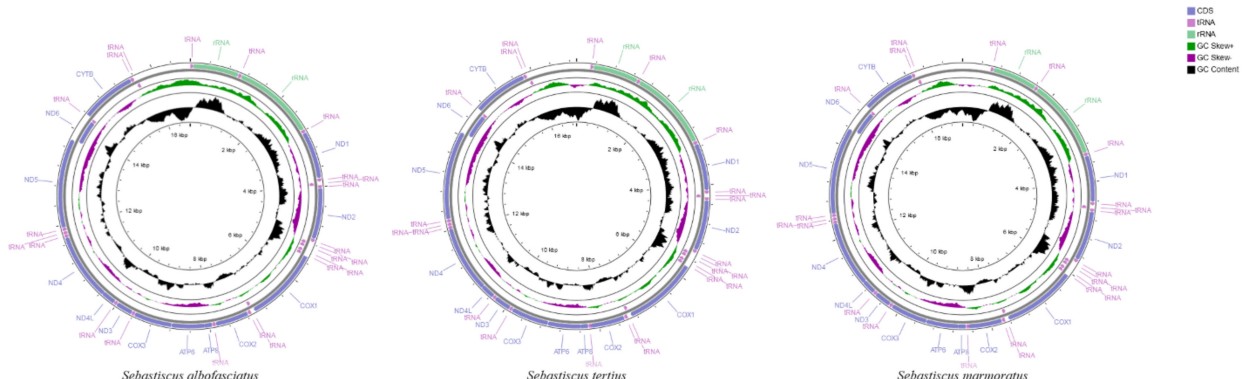

**Figure 5.** Mitochondrial genome maps of *S. tertius*, *S. albofasciatus*, and *S. marmoratus*. Genes encoded on the heavy or light strands are shown outside or inside the circular gene map, respectively.

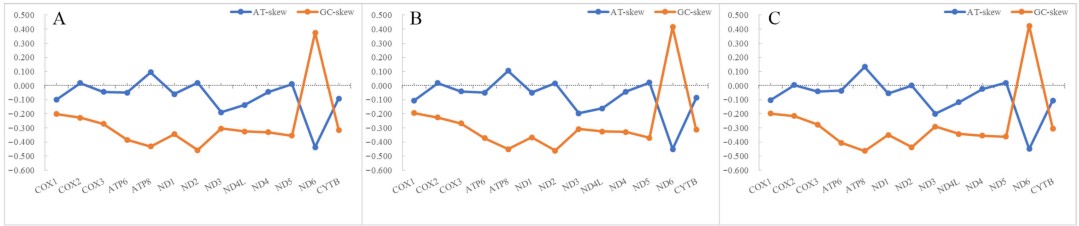

**Figure 6.** The nucleotide skewness of three species of *Sebastiscus*. (**A**) *S. tertius*; (**B**) *S. albofasciatus*; (**C**) *S. marmoratus*. The incomplete T–/TA- of the stop codon is not included.

Phylogenetic relationships were reconstructed based on the sequences of 13 PCGs of 27 mitogenomes using NJ and ML methods. The phylogenetic trees constructed by two methods were consistent with high intermediate bootstrap values and the topological structure of the two phylogenetic trees was entirely the same (Figure 7). Moreover, six *Sebastiscus* sequences formed a monophyletic group either in the NJ or ML analyses. *S. tertius* formed a sister group with *S. albofasciatus* and together had a sister relationship with *S. marmoratus*, which is consistent with previous studies [8].

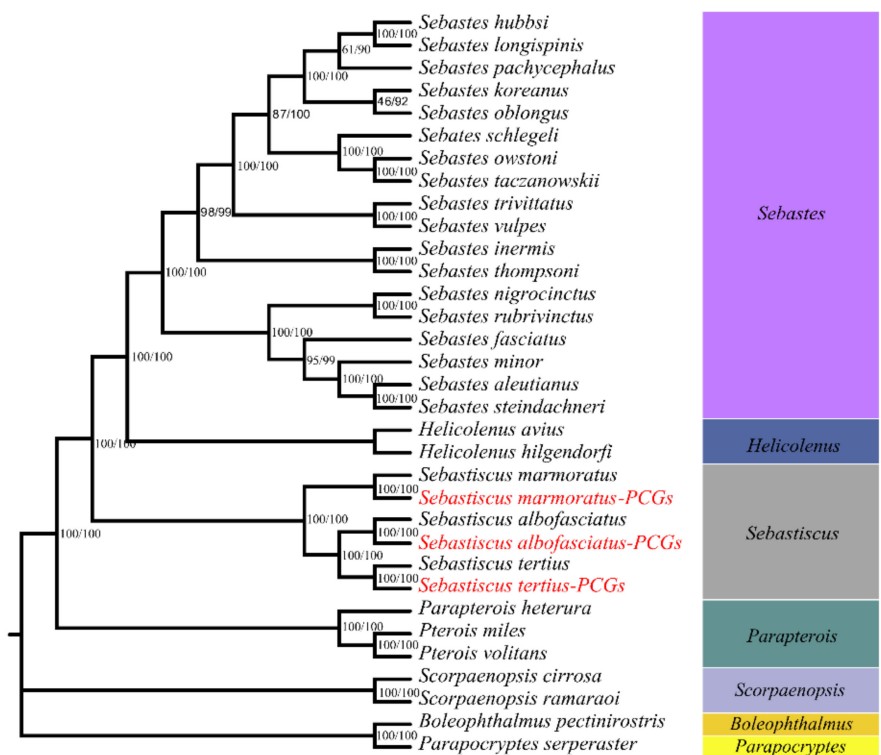

**Figure 7.** Phylogenetic tree of 25 Scorpaenidae sequences constructed by neighbor-joining (NJ) and maximum likelihood (ML) methods based on concatenated sequences of 13 PCGs. *Boleophthalmus pectinirostris* and *Parapocryptes serperaster* were used as the outgroup. The species in red Latin name indicates the sequences generated in this study.

## 3. Discussion

Prior to this study, the genetic information about *Sebastiscus* was only limited to *S. marmoratus* and the mitochondrial genome sequences of other *Sebastiscus* species [8,12,14,15]. In recent years, genomic technology has developed rapidly with the application of NGS technology, which provides an affordable way to solve a wide range of questions [16–18]. As a result, more and more non-model marine fish species have been sequenced and studied by genome survey sequencing [19–21]. In the present study, we reported the genome survey of all *Sebastiscus* species by the whole genome sequencing for the first time. Moreover, the microsatellite motifs were identified, and the complete mitochondrial genomes were also assembled, all of which could provide a valid reference in future genome and molecular marker research for marine fishes.

The k-mer analyses suggested that the genome sizes of *Sebastiscus* were 802.49 Mb (*S. albofasciatus*), 786.79 Mb (*S. tertius*), and 776.00 Mb (*S. marmoratus*), respectively. From the results, the genome size of *S. marmoratus* (776.00 Mb, k = 19) was lower than the previous study (796.25 Mb, k = 21) [12]. This might be explained by the accuracy of different parameters or the intraspecific genome size variation, which caused discrepancies in results [10,22,23]. Besides the variation in genome size, both the heterozygosity ratio and repeat ratio in *S. marmoratus* were relatively larger than other *Sebastiscus* species. The higher

heterozygosity of *S. marmoratus* also suggested that *S. marmoratus* had higher population genetic variability than the other two. The difference between the former and the latter may account for one of the genetic differences between species. In a word, the genome survey analyses in the present study provided fundamental and valuable information in genomics and molecular biology of *Sebastiscus*.

For the genome assembly, lower heterozygosity showed that all the genomes of three *Sebastiscus* were relatively simple and all *Sebastiscus* species would be preferable for the development of a draft genome in future studies [24,25]. It is consistent with previous research about *Sebastiscus* [12]. This is the first genome survey comparison of all species in *Sebastiscus* genus and the assembled genome sequences in this study can be useful for further genomic studies.

Molecular markers are an ideal form of genetic marker. In addition to facilitating detection, multiple allele polymorphism, and codominant inheritance, molecular markers also possess advantages that are not found in rapid fragment length polymorphism (RFLP) and amplified fragment length polymorphism (AFLP) markers [26]. All *Sebastiscus* genomes appeared capable of developing tremendous SSR markers, which would help to solve the problem that SSR markers only could be derived from the genome data in *S. marmoratus* [10]. The frequency of SSR repetitions decreases exponentially with the length of SSR repetitions, as longer mutations have higher mutation rates [27]. This is consistent with the result that the number of repetitions is inversely proportional to the length of repetitions reported by Chen et al. (2010) [28]. Moreover, the number and type of SSR markers were lower than those of other *Sebastiscus* species, especially in tetranucleotide microsatellite motifs of *S. marmoratus*. It might be due to *S. marmoratus* producing the genetic mutations during evolution, which eventually formed the new *Sebastiscus* species and generated the genome-wide difference. Statistical analysis of the differences in the quantity and types of SSRs in *Sebastiscus* and an initial exploration of the genome data provided a foundation for the further construction of high-density genetic maps of rockfishes.

The whole genome sequencing data also included extranuclear genome like mitochondria [29]. The mitochondrial genomes of three *Sebastiscus* species were assembled from the whole genome sequencing data by using Mitofinder software. The result of phylogenetic tree based on 13 PCGs also revealed the accuracy of the mitochondrial genome assemblies by the method. Mitochondrial genomes have become a powerful molecule marker for species classification, population genetics, molecular systematic geography, molecular ecology, and other fields [30–34]. However, short mitochondrial gene fragments still harbor some limitations in discussing and resolving more complicated phylogenetic relationships in many fish lineages [35]. For these limitations, the longer DNA sequences liked protein-coding genes in complete mitochondrial genomes which have additional informative sites will have better ways to solve these higher-level relationships and deeper branches thoroughly [36]. The same topology generated by different methods proved it. We suggest that a cost-effective method to assemble mitochondrial genomes should be widely used in future genome survey studies.

## 4. Materials and Methods

### 4.1. Sample Collection and Preservation

The samples of *S. tertius* and *S. marmoratus* were collected using hook-and-line fishing from the coastal waters of Taizhou in China (coordinates: 28.54° N, 121.64° E) during December 2020 and Qingdao in China (coordinates: 35.76° N, 120.20° E) during May 2018, and *S. albofasciatus* originated from Kozagawa in Japan (coordinates: 33.41° N, 135.75° E) during June 2019, respectively. All samples were identified based on morphological characteristics [37] and one random individual was chosen per species—for genome sequencing. Muscle tissues were stored in 95% ethanol at −80 °C for further study.

### 4.2. Genome Survey Sequencing

Total genomic DNA was extracted using a standard phenol-chloroform method for muscle tissue. DNA was treated with RNase A to produce pure, RNA-free DNA. Two paired-end DNA libraries were constructed with insert size of 350 bp, and then sequenced using the Illumina HiSeq 4000 platform following the manufacturer's protocol. The library construction and sequencing were performed at Biomarker Technologies in Beijing. The whole genome sequencing data were deposited in the Short Read Archive (SRA) database (http://www.ncbi.nlm.nih.gov/sra/ accessed on 18 July 2021) under accession numbers PRJNA746673, PRJNA746685, and PRJNA722703, respectively.

### 4.3. K-Mer Analysis and Genome Assembly

The genomic size and heterozygosity were estimated using k-mer analysis method. After removing low quality reads, all clean data were used to perform k-mer analysis, using the 350 bp library data and K = 19 to build the profile. Based on the results of the k-mer analysis, information on peak depth and the number of predicted best k-mer were obtained and used to estimate the size of the genome. Its relationship was expressed by using the following algorithm [29]: genome size = k-mer_num/peak_depth, where k-mer_num is the total number of predicted best k-mer, and peak_depth is the expected value of the k-mer depth. Additionally, the heterozygosity ratio and repeat sequence ratio were estimated following the description in [26], based on the k-mer analysis. K-mer analyses were performed using software GCE v1.0.0 [38]. The clean reads were assembled into contigs in software SOAPdenovo v2.01 [39] with a k-mer of 41 by applying the de Bruijn graph structure. The paired-end information was then used to join the unique contigs into scaffolds.

### 4.4. Microsatellite Identification

The number and types of microsatellites can be identified by analyzing the genome sequence. The software MIcroSAtellite (MISA, http://pgrc.ipk-gatersleben.de/misa/ accessed on 20 July 2021) was used to identify microsatellite motifs in the de novo draft genome sequences [40,41].

### 4.5. Mitogenome Assembly and Phylogenetic Analysis

The filtered clean data were assembled and mapped to complete mitogenome sequence using Mitofinder [42]. All complete mitogenomes were preliminarily annotated and the mitochondrial genome map was drawn by Mitofish (https://mitofish.aori.u-tokyo.ac.jp, accessed on 24 July 2021) [43,44].

In order to discuss and verify the accuracy of the mitogenome sequence, mitogenomes of previously sequenced Scorpaenidae (22 species) and Gobiidae (2 species, the outgroup taxon) were used in the phylogenetic analysis. We used the nucleotide sequences of the 13 protein-coding genes (PCGs, including ND1, ND2, COI, COII, ATP8, ATP6, COIII, ND3, ND4L, ND4, ND5, ND6, and CYTB) as the dataset to construct the phylogenetic tree. Sequences were aligned using SeqMan from DNAStar software (USA). The optimal model for nucleotide sequences was estimated by MEGA X [45]. The GTR + G + I model was considered to be the best one for phylogenetic tree construction since it captured the minimum values of Bayesian Information Criterion (BIC) and Akaike Information Criterion (AIC). The Maximum Likelihood (ML) phylogenetic tree was constructed by MEGA X software with 1000 replicates of bootstrap.

## 5. Conclusions

In the present study, the first genome survey study of all *Sebastiscus* species was performed based on whole genome sequencing data of three different individuals. The genome sizes of *Sebastiscus* were 802.49 Mb, 786.79 Mb, and 776.00 Mb, respectively. Then, comparative analyses among the three species were also investigated to reveal possible interspecific differences. The heterozygosity, microsatellite, and other data all showed

that the external morphology of the *S. marmoratus* and *S. tertius* is more similar, but *S. tertius* is more closely related to *S. albofasciatus* at the genetic level. This will provide a meaningful reference for future research on the origin and evolution of the *Sebastiscus* genus. Moreover, the mitochondrial genome assembly and phylogenetic analysis were integrated into our genome survey study. The result of the phylogenetic tree also proved the accuracy of the mitochondrial genome assemblies through the method. This can be used to provide an important genome resource for further studies of *Sebastiscus* species. In addition, we suggested that further studies should be continued by high-quality whole genome sequences of *Sebastiscus* based on the combination of "Illumina + PacBio + Hi-C" techniques, to provide valuable information for genomic and evolutionary biology studies.

**Supplementary Materials:** The following are available online at https://www.mdpi.com/article/10.3390/cimb43030141/s1, Table S1: Base composition of *Sebastiscus* mitochondrial genomes.

**Author Contributions:** Data curation, C.J. and T.G.; formal analysis, C.J. and T.Y. (Tianyan Yang); methodology, T.G.; writing—original draft, C.J.; writing—review and editing, T.Y. (Tianyan Yang), T.Y. (Takashi Yanagimoto), and T.G. All authors have read and agreed to the published version of the manuscript.

**Funding:** This study was supported by the National Key R&D Program of China (2019YFD0901303) and National Natural Science Foundation of China (No. 41776171).

**Institutional Review Board Statement:** Ethical review and approval were waived for this study, due to the samples were dead before retrieved.

**Informed Consent Statement:** Not applicable.

**Data Availability Statement:** The data presented in this study are openly available in Short Read Archive (SRA) database, reference number [PRJNA746673, PRJNA746685, and PRJNA722703].

**Acknowledgments:** We sincerely thank the reviewers for their critique and suggestions. Besides, we sincerely thank Qi Liu for the data analysis guidance.

**Conflicts of Interest:** The authors have declared that no competing interests exist.

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
