# Peer review of "Comprehensive Draft Genome Analyses of Three Rockfishes (Scorpaeniformes, Sebastiscus) via Genome Survey Sequencing"

_cimb, doi:10.3390/cimb43030141_

Round 1
Reviewer 1 Report
The manuscript entitled “A New Insight into Comprehensive Draft Genome analyses of three Rockfishes (Scorpaeniforme, Sebastiscus) via Genome Survey Sequencing” is a genome description of three fish species from the Pacific belonging to the genus Sebastiscus.
It is a survey well presented and developed. The fact that this manuscript is concise is good, without useless sentences. But sometimes some more explainations would be welcome.
- E.g.: the description of the species is very short, “interesting and commercial species”. Why they are interesting? Are they eaten? Sold for aquariums? Also distributions of single species would be useful, and to know if these distributions are overlapped. Are there any subspecies described for these species?
- The title can be shortened in “Comprehensive Draft Genome analyses of three Rockfishes (Scorpaeniforme, Sebastiscus) via Genome Survey Sequencing”
- Previous genetic studies on these species were focused on which markers? E.g. Mitochondrial gene sequences, microsatellites, nuclear segments?
- I didn’t found an evolutionary explanation in the text for the longer genome of one species in comparison to the other two. What is the possible cause of that?
- Please check some language oversights (line 81 “there are…. were identified”).
- Paragraph 4.1: how many samples for each species have you analysed? Please specify here. If just one individual per species was analysed (see line 229), I think the authors should highlight this weakeness of the study, because it can be possible/probable to have a different picture from several animals of the same species.
- The authors could discuss in detail how new microsatellites discovery and description can be useful for population studies, identification of single individuals and definition of Units for Conservation or Management of the species. Some sentences in discussion and conclusions are too much concise and vague.
- Similarly, how does this genome assembly prove beneficial and valuable for the management of these and other fish species and/or how does this study enhance the current methodological scenario?
When this manuscript is improved in these points, which would also make it more appealing to a wider audience, I think it may be ready for publication.
Author Response
Dear reviewer,
We are deeply grateful to your critical comments concerning our manuscript entitled “A New Insight into Comprehensive Draft Genome analyses of three Rockfishes (Scorpaeniforme, Sebastiscus) via Genome Survey Sequencing” (cimb-1411902). The useful comments are very helpful for improving and revising our manuscript. Based on these comments, careful modifications have been made on the revised manuscript. Please see the attachment which includes details of all changes about this manuscript. Other inconveniences in the article to make changes, we also carried out detailed explanations in the "response to reviewer".
We are looking forward to hearing from you soon for a favorable decision. Thank you again for your time and consideration. If you have any questions, please don’t hesitate to contact us.
Yours sincerely,
Tianxiang Gao

Reviewer 2 Report
The proposed manuscript introduces a genome survey of three Sebastiscus species. Unfortunately the proposed manuscript is very descriptive and is presented in an ordinary way. I did not find a convincing impact of the study and have found some fundamental issues. In addition, extensive English revision is needed and chosen abbreviations are not clearly explained for better understanding.
Serious flaws: explanation of fundamental terms and abbreviations is missing (e.g., clean data, Q20, Q30, PCGs), inappropriate alternation of present and past tense within the text, missing Figure 2, not fully described phylogenetic tree - I am not sure what genes were used for evolutionary relationships and why.
Overall, presented outputs do not seem to have sufficient content for publication in the CIMB scientific journal. I suggest combining these submitted results together with further included studies of Sebastiscus species exactly as the authors have outlined (lines 22-23, 236-237), or send the manuscript to another specific journal.
Author Response

(The authors gave the same response as above.)

Reviewer 3 Report
Manuscript ID: cimb-1411902The article presents the results of whole genome sequencing along with identification of microsatellite motifs and mitogenome assembly in three rockfishes of the genus Sebasticus, particularly S. albofasciatus, S. marmoratus and S. tertius. Below are comments to the reviewed manuscript:
Comment 1
Lines 29-30: The authors of scientific names of particular fish species should be mentioned directly in the text (i.e. the generic and specific name, followed by the correct authority of the taxon), and not only in the reference list. Do authors really have access to articles of Lacepede, and Cuvier & Valenciennes published in 1802 and 1829, respectively? Is the citation stated for Sebasticus tertius correct? Please see “FishBase database”, a global species database of fish species.
Divide the sentence in the Lines 29-32 into two individual sentences. Consider reformulating the sentences in the Lines 34-41 since all of them starts with “Although, Therefore, However”.
Comment 2
Lines 42-45: Authors provided incorrect references for this statement, especially for its last part „....widely applied in genomic and evolutionary studies of MARINE FISH species [references nos. 7-11].“ The following three articles do not dealt with marine fish species:
Reference no. 7 – Jennings et al. 2011: Seven microsatellite-enriched, barcoded genomic libraries were prepared from two conifer trees and five birds.
Reference no. 8 – Králová-Hromadová et al. 2015: Microsatellite markers were developed for monozoic fish tapeworm Caryophyllaeus laticeps (Cestoda: Caryophyllidea) which has been found in freshwater fish species such as Abramis brama, Ballerus sapa, Cyprinus carpio etc.
Reference no. 9 – Lu et al. 2016: The genome survey of the rose Rosa roxburghii using NGS was conducted to investigate and provide a genomic resource of this species.
Please cite the adequate reference or reformulate the sentence in the Lines 42-45. In general, the introduction should undergo revision and more complex narrative around should be provided.
Comment 3
It is not standard to present section “Material and Methods” after the “Discussion”. The recent articles published in the Current Issues in Molecular Biology DO NOT followed this strategy.
Comment 4
The one of the weakness of the study is, according to my opinion, the fact that the results have not been discussed satisfactory. Authors just briefly mentioned four previous articles (ref no. 6, 10, 12, 13) without deeper insight, comparison and meaningful discussion. The effort to compare genome size of one male adult Sebasticus marmoratus recently estimated by Xu et al. (2020) with the current results was obvious; however, it was not presented in a satisfactory manner. I would recommend to revalue the relevance of content and to improve these issues in the revised manuscript.
Besides, the repetition of the statement “….results could be/can be/should be useful in the future genomic studies/genome survey/further studies” has been probably overlooked (see lines 144-147; 160-162; 174-175). Avoid repeating using of “in this study”.
Comment 5
Authors did not provide any specific conclusion and more detailed future perspectives of their study. Statements provided in the conclusion are simply repeated from the abstract and presented in almost the same way. Moreover, the formulation “in this study” is used three times in the conclusion.
Other comments
- Figure 3 is missing.
- Reformulate the sentence starting in the Line 81 and 93, respectively.
- Do not start the sentences with “And” – see Lines 82, 93, 148, 152
- Explain the meaning of squares in different colours with particular numbers in Figure4 A-C, Figure4 D-F, Figure4 G-I.
- Line 181: replace “was come from” with “originated from”
- Reformulate sentences in the Lines 127-130.
- Sample collection – specify the exact number of samples.
- I would recommend English editing and proofreading by native speaker.
Author Response

(The authors gave the same response as above.)

Round 2
Reviewer 2 Report
The re-submitted manuscript “Comprehensive Draft Genome Analyses of three Rockfishes (Scorpaeniformes, sevastiscus) via Genome Survey Sequencing“ was improved and if the editor agrees, that the proposed manuscript meets the requirements of the CIMB journal, I have no problems with acceptance after major revision. Manuscript is well written, presented in an intelligible fashion, and the structure is well designed. Despite the fact that I do not feel qualified to judge about the English language and style, there are some grammatical errors. I recommend that the manuscript be read by native English-speaking person. In the previous version, Fig 1 was missing (neither Fig 2 nor Fig 3) which was fixed in the current manuscript version. Overall, I suggest major revision is needed.
- One example where English grammar has not been correctly used: line 117: “GC content of clean data was shown in Table 1 and Fig 1”. Line 119: “distribution derived from sequencing reads is plotted in Fig 2”. In the first case, the label of Fig 1 in the plain text is expressed in the present simple. The label of Fig 2 is in the past simple. Two cases but one story thus the same grammar should be used.
- The typo in manuscript title: “Scorpaeniforme” instead of “Scorpaeniformes”!!
- In the manuscript citation in the left column (line 24-25?), authors have used “Int. J. Mol. Sci.”. It seems that the manuscript was submitted to another journal then rejected and sent to CIMB. Am I right??
Author Response
Dear reviewer,
We are deeply grateful again to your critical comments concerning our manuscript entitled “Comprehensive Draft Genome analyses of three Rockfishes (Scorpaeniformes, Sebastiscus) via Genome Survey Sequencing” (cimb-1411902). The useful comments are very helpful for improving and revising our manuscript. Based on these comments, careful modifications have been made on the revised manuscript. Please see the attachment which includes details of all changes about this manuscript.
We are looking forward to hearing from you soon for a favorable decision. Thank you again for your time and consideration. If you have any questions, please don’t hesitate to contact us.
Yours sincerely,
Tianxiang Gao

Round 3
Reviewer 2 Report
Authors have followed all my recommendations, and the proposed manuscript can be accepted. I do appreciate to see published version soon.